# DEMO-ICL: IN-CONTEXT LEARNING FOR PROCEDURAL VIDEO KNOWLEDGE ACQUISITION

## ABSTRACT

Despite the growing video understanding capabilities of recent Multimodal Large Language Models (MLLMs), existing video benchmarks primarily assess understanding based on models' static, internal knowledge, rather than their ability to learn and adapt from dynamic and new contexts with minimal examples. To bridge this gap, we present **Demo-driven Video In-Context Learning**, a novel task focused on learning from in-context demonstrations to answer the target videos. Alongside this, we propose **Demo-ICL-Bench**, a challenging benchmark specifically designed to evaluate demo-driven video in-context learning capabilities. The Demo-ICL-Bench is constructed using 1200 instructional YouTube videos with questions, from which two types of demonstrations are derived: summarizing video subtitles for text demonstration or directly using a corresponding instructional video as video demonstration. To effectively tackle this new challenge, we develop **Demo-ICL**, an MLLM with two-stage training strategies: video-supervised fine-tuning and information-assisted direct preference optimization, jointly enhancing the model's ability to learn from in-context examples. Extensive experiments with state-of-the-art MLLMs confirm the challenges of Demo-ICL-Bench, demonstrate the effectiveness of Demo-ICL, thereby unveiling future research directions.

## 1 INTRODUCTION

Video understanding has been a long-standing and significant challenge in computer vision. Recent Multimodal Large Language Models (MLLMs) (Alayrac et al., 2022; Awadalla et al., 2023; Huang et al., 2023; Zhao et al., 2023; Peng et al., 2023) have achieved significant progress in video benchmarks, expanding capabilities from short-clip recognition (Fabian Caba Heilbron & Niebles, 2015; Mangalam et al., 2023; Goyal et al., 2017) to the analysis of long videos (Chandrasegaran et al., 2024), and handling videos from daily life videos (Lei et al., 2019; Mangalam et al., 2023) to instructional videos (Miech et al., 2019; Tang et al., 2019; Zhukov et al., 2019).

However, existing video benchmarks typically pose questions that rely on either internal, general pre-trained knowledge (e.g., asking "what is a whisk?") or simple facts observable within the specific target video being evaluated (e.g., "where is the whisk?"). This is fundamentally different from the more challenging scenario where a model must learn a new process or skill from demonstrations (e.g., a video tutorial that teaches the model how to cook Mexican Rice) and then apply that learned knowledge to answer questions based on a new, related target video sequence. This scenario reflects human learning and is crucial for downstream applications like robotics, where robots can learn from demonstrations to tackle new tasks. For instance, in Fig. 1, the model is required to watch a video showing only the initial step of heating oil for Mexican Rice and being asked "what should you do next?" based on *in-context* text instructions or video demonstrations. This question requires knowing the specific sequence of steps for this particular version of Mexican Rice that the model is presumably meant to understand or follow, such as based on the in-context video demonstrations.

To better encourage models to learn and adapt new skills from context, we propose a challenging video understanding task called **Demo-driven Video In-Context Learning (Demo-driven ICL)**. Our task embodies this by presenting target videos and questions alongside *in-context* text guidelines or video demonstrations. As shown in Fig. 1, Demo-driven ICL has three sub-settings: (1) text-demo in-context learning, (2) video-demo in-context learning, and (3) demonstration selection. These questions explicitly require models to use the knowledge provided within in-context examples, rather than just their static internal knowledge. A key difference between Demo-driven ICL and

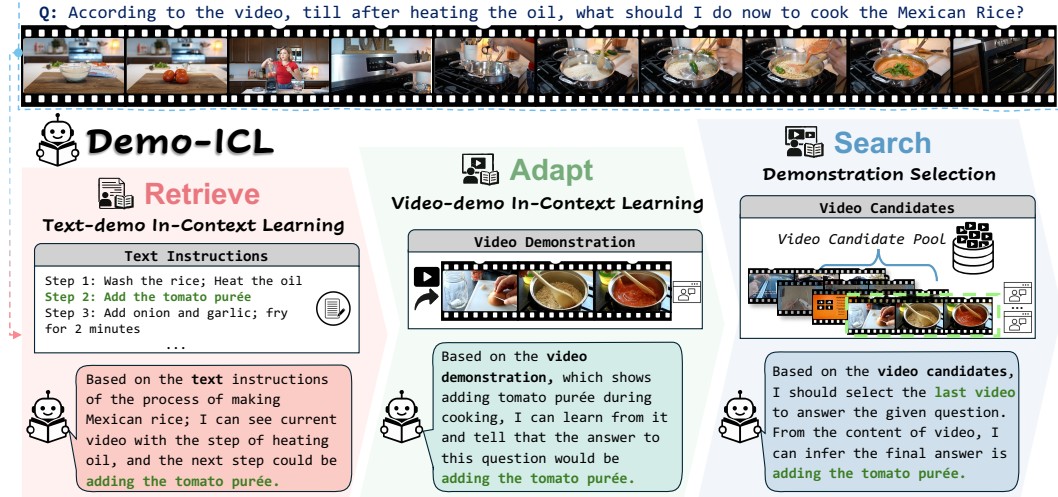

Figure 1: **Overview of the Demo-driven Video In-Context Learning Task** with three distinct settings: (1) *Text-demo in-context learning*, where text instructions act as the demonstrations; (2) *Video-demo in-context learning*, where a video demonstration acts as the reference; and (3) *Demonstration Selection*, which requires identifying the most relevant video demonstrations among the video candidate pool and using them to guide in-context learning.

previous in-context learning paradigms lies in the data modality and interaction type: our in-context demonstrations could be videos, and the task involves choosing from the video candidate pool to identify suitable in-context examples by the model itself. The Demo-driven ICL tasks mirror how one might learn a complex skill, such as cooking, by searching related video demonstrations and watching while also consulting supplementary visual or textual guides.

To evaluate the proposed Demo-driven ICL task, we present **Demo-ICL-Bench**, a benchmark that consists of text/video demonstrations, target videos, questions, and answers. We collected instructional YouTube videos from the dataset by (Miech et al., 2019), ensuring subtitles and timestamps. Subsequently, we used an LLM to summarize these subtitles, generating text demonstrations to serve as in-context examples. Additionally, we employ video search ranking methods and an LLM to identify and select videos similar to the target video to serve as in-context video demonstrations. We also construct a video candidate pool for the model to select and learn, in order to mimic real-world scenarios Demo-ICL-Bench is complex and challenging: answering every question demands an accurate understanding of the demonstrations, resulting in frontier models like Gemini-2.5-Pro achieving merely 46.6% and 32.0% accuracy when processing text and video demonstrations, respectively.

To further address the Demo-driven ICL task, we further present a **Demo-ICL** model with two-stage training strategies: video supervised fine-tuning, and information-assisted Direct Preference Optimization (DPO) for demo-driven video in-context learning. We design an information-assisted DPO data generation pipeline to produce high-quality chosen responses by simplifying the task with additional contextual information. Demo-ICL outperforms existing MLLMs on various benchmarks, such as the proposed Demo-driven ICL, VideoMMMU for video knowledge acquisition (Hu et al., 2025), and VideoMME (Fu et al., 2024a) for general video understanding.

Our key contributions are: **(i) New Challenging Tasks**: We design three Demo-driven Video In-Context Learning (Demo-driven ICL) tasks, which enable models to answer questions by learning from text or video demonstrations, representing a significant step towards more human-like learning and decision-making processes in video understanding tasks. **(ii) New Benchmark and Evaluation**: We establish a new Demo-ICL-Bench that is specifically designed for evaluating demo-driven video in-context learning capabilities. Based on Demo-ICL-Bench, we conduct comprehensive evaluations of cutting-edge baselines, showcasing various challenges of our proposed task. **(iii) New Demo-ICL Model**: We present a new model, Demo-ICL, along with a customized two-stage training strategy that enhances a model's ability to learn and adapt from in-context demonstrations. Compared with SOTA models, Demo-ICL shows competitive performance across existing benchmarks (Fu et al., 2024a; Wu et al., 2024; Hu et al., 2025), which demonstrate its superior video comprehension and in-context knowledge acquisition capabilities.

## 2  RELATED WORK

**Multimodal Video Understanding for Knowledge Acquisition.** Multimodal video understanding has increasingly shifted from low-level perception toward knowledge acquisition, which is the ability to extract, structure, and apply information from complex instructional video data. Large-scale instructional datasets have been created and are central to knowledge extraction. HowTo100M (Miech et al., 2019) introduced 1.2 million narrated videos with 136 million clip–caption pairs for procedure recognition and cross-task transfer. Other datasets (Tang et al., 2019; Zhukov et al., 2019) provide fine-grained task annotations or support weakly supervised step parsing across diverse procedures. More recently, Video-MMMU (Hu et al., 2025) and VideoMathQA (Rasheed et al., 2025) evaluate models' capability to learn from educational videos, shifting emphasis from perception to knowledge uptake and application. To further benchmark the video understanding task for knowledge acquisition in real-world applications, we introduce Demo-driven Video In-Context Learning task and propose Demo-ICL-Bench, where models should acquire new concepts from given demonstrations. Please refer to Table 6 in the Appendix to see a detailed comparison between our benchmark and others.

**Multimodal In-Context Learning.** In-context learning (ICL) enables models to perform new tasks by conditioning on a few examples at inference time. Initially developed for large language models (LLMs), ICL has been extended to multimodal settings and shows consistent performance gains across both language and image tasks (Brown et al., 2020; Min et al., 2021; Luo et al., 2024; Zhou et al., 2024b). However, video-based ICL still remains underexplored, where current video MLLMs (Lin et al., 2024a; Maaz et al., 2024; Zhang et al., 2025b) mainly emphasize zero-shot performance through curated video instruction datasets, to equip models with open-ended QA, captioning, and dialog capabilities. Recent research has begun to address the context challenges for video tasks. For example, chain-of-thought methods for video (Wang et al., 2024b; Han et al., 2025; Arnab et al., 2025) encourage stepwise evidence aggregation and explicit explanation, while some retrieval-based methods (Ren et al., 2025; Tevissen et al., 2024) establish a new paradigm of retrieving video moments and ground answers in cited segments. However, aforementioned works mainly use the context as references, rather than adapting to and learning from the context. In our work, we address this gap by introducing Demo-driven ICL task on instructional video datasets, supported by an optimized training pipeline, to enhance the model's capability to learn from the in-context demonstrations.

## 3  DEMO-ICL: LEARNING FROM IN-CONTEXT VIDEO DEMONSTRATIONS

In this section, we first provide a detailed overview of the proposed Demo-driven Video In-Context Learning tasks, explaining both the task formation and dataset construction. Section 3.1 defines and categorizes these tasks, while Section 3.2 outlines the carefully designed dataset construction process used to generate our training and validation datasets. Finally, we demonstrate how we train our model to achieve demo-driven video knowledge acquisition in Section 3.3.

### 3.1  DEMO-DRIVEN VIDEO IN-CONTEXT LEARNING

Learning from demonstrations and imitating actions are crucial skills for humans when acquiring new abilities. Such capabilities enable individuals to rapidly master novel tasks from only a handful of examples, thereby supporting efficient adaptation and facilitating lifelong learning. In contrast, contemporary video models largely depend on supervised fine-tuning to acquire task-specific capabilities introduced by previous benchmarks, neglecting the importance of learning from in-context examples and evaluating such capabilities for achieving human-like performance. Additionally, humans often learn new tasks incrementally. This alignment underscores the need for models that support procedural video knowledge acquisition, enabling the incremental internalization and generalization of task procedures in a human-like manner.

To address such problems, we propose a new set of three tasks called **Demo-driven Video In-Context Learning**. These three tasks are designed to evaluate the model's ability to learn from in-context demonstrations. Given an instructional video $V_D$ or text demonstration $T_D$, the model must first interpret the example to understand how a task should be completed. It is then presented with a test video $V_{Test}$, and its ability to transfer knowledge from the demonstration is assessed by predicting subsequent steps of an action $A_{[t_1,t_2]}$ based on the demonstration and the available context $V_{Test}[0, t_1]$. Depending on the format and source of demonstrations, we define three distinct tasks:

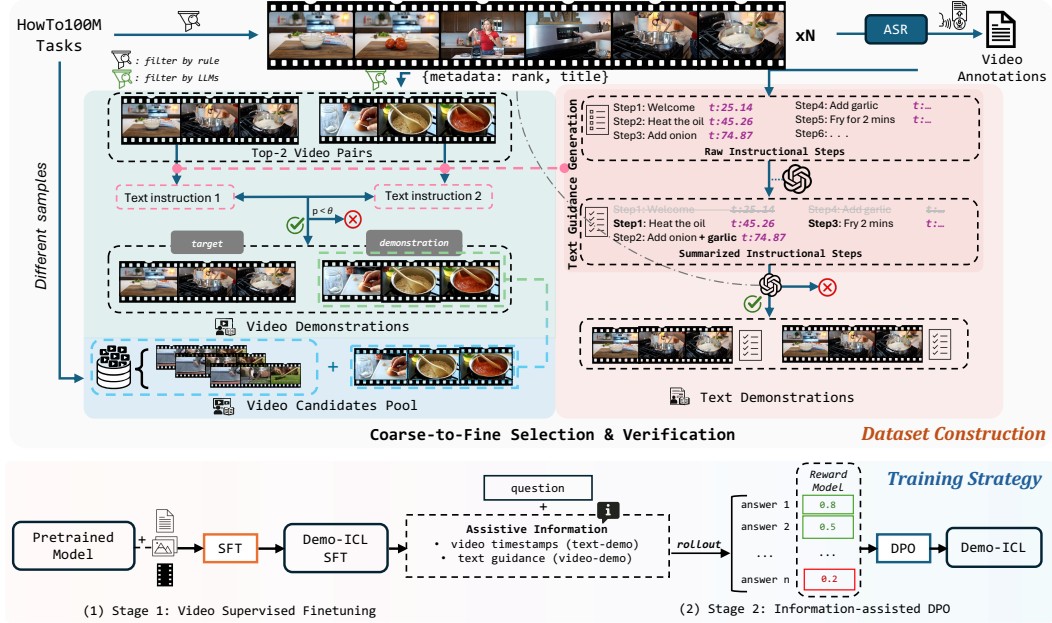

Figure 2: **Overview of Data Construction and Training Strategy.** (i) illustrates our coarse-to-fine dataset collection pipeline (Section 3.2); (ii) presents the tailored two-stage training strategy for training the Demo-ICL model (Section 3.3).

**1) Text-demo In-Context Learning:** The model answers questions from an input video (e.g., "what should I do now to cook the Mexican Rice") by retrieving information from the corresponding **textual instructions** (e.g., "Step 1: Wash the rice; Step 2: Add the tomato; Step 3: . . . "), which serve as the demonstration. For example: "Based on the *text instructions*, the current video corresponds to Step 2. Therefore, the next step is Step 3: add the onion."

**2) Video-demo In-Context Learning**: The model answers questions about a target video by conditioning on a provided **video demonstration** of a similar task, and using the demonstration as an in-context exemplar. For instance: "Given the *video demonstration*, the target clip aligns with Step 2; therefore, the next step is Step 3: add the onion."

**3) Demonstration Selection**: The model is given the input video and a pool of video candidates (e.g., a pool containing "Mexican rice," "fried rice," and "pasta,"). The model must first **select the most relevant demonstration** (e.g., "Mexican rice") from the pool and then use it to answer the question, simulating a scenario where a perfectly aligned demonstration is not provided.

Collectively, these three tasks constitute a systematic and comprehensive framework for demo-driven video in-context learning. They underscore distinct model capabilities, ranging from textual retrieval to demo-based knowledge extraction and adaptation, and correspond to successive stages of development, spanning from idealized oracle settings to practical real-world scenarios.

## 3.2 DATASET CONSTRUCTION

In this section, we introduce a comprehensive data generation pipeline to support the proposed demo-driven video in-context learning task. The pipeline emphasizes four key qualities: ensuring that the video content is informative, the textual demonstration is precise, the video demonstrations are contextually relevant, and the generated questions remain answerable. To meet these requirements, we develop a structured step-by-step process (see Fig. 2 (i)) and detail each component below.

**Video Collection and Annotation.** We use video data from HowTo100M (Miech et al., 2019), a large-scale corpus of narrated YouTube instructional videos designed for complex tasks. This dataset is particularly suitable for demo-driven video in-context learning, as it allows models to acquire procedural knowledge from step-by-step demonstrations. With over 100 million clips covering 23,000 activities, HowTo100M provides diverse and extensive instructional material for building

our benchmark. After selecting this source, we first filter videos based on video length, language, and title availability. To obtain high-quality annotations, we use ASR outputs as they offer more detailed descriptions of demonstrated activities compared with video captions. Specifically, we use annotations from HTM-AA (Han et al., 2022), which employs WhisperX (Bain et al., 2023) to generate sentence and word-level timestamps.

**Text Demonstration Generation.** We generate textual demonstrations for each video using a coarse-to-fine pipeline that produces step-by-step instructions. First, an LLM summarizes the ASR transcripts into a sequence of clips, identifying the instructional steps needed to complete the task. Next, a second LLM filters out irrelevant steps, retaining valid steps while merging redundant ones into the nearest relevant segment to preserve continuity. This process yields a coherent, task-focused sequence of instructions. Finally, an MLLM refines the demonstrations by jointly considering the step descriptions and corresponding video clips, ensuring contextual accuracy and close alignment with the depicted actions. Through this multi-stage refinement, we obtain precise and reliable textual guidance that captures both the procedural structure of the task and its visual grounding.

**Video Demonstration Selection.** To enable video-based in-context learning, we construct pairs of videos that illustrate similar tasks. These paired demonstrations serve as explicit visual guidance, allowing the model to observe alternative executions and acquire procedural knowledge from reliable examples. Pair selection follows a coarse-to-fine process. We first leverage metadata from HowTo100M, which provides YouTube search rankings for specific tasks, and discard tasks with few relevant videos to ensure pairing quality. Next, an LLM evaluates the titles of top-ranked videos for each task and selects the two with the highest semantic similarity. To further validate these candidates, we generate textual instructions for both videos using the previous pipeline, after which the LLM compares the instructions to confirm that the videos indeed demonstrate similar tasks that can be transferred. This process ensures both accuracy and reliability of the selected video guidance.

**Demo-driven Question Generation.** With curated videos and corresponding instructions, we construct questions to evaluate demo-driven video in-context learning. For text-based in-context learning, we exclude tasks with fewer than six steps to ensure sufficient complexity. From each valid sequence, one intermediate step (excluding the first and last) is randomly selected, and a question is generated from this step. The model is then required to predict the next action.

For video-based in-context learning, the goal is to test whether models can effectively leverage visual demonstrations. To this end, we employ an LLM to analyze the generated instructions for paired videos, determining whether they represent comparable tasks suitable for question generation. If validated, the LLM identifies the target step and its corresponding timestamp within the pair. Human annotators then assess the generated questions, focusing on whether the visual demonstrations provide meaningful contextual evidence for answering. To avoid trivial cases, we further filter out highly similar video pairs, ensuring the task meaningfully tests the model's adaptability and generalization.

For the demonstration selection task, the validated video pairs are treated as ground truth and augmented with carefully chosen irrelevant videos. This construction requires models to distinguish informative demonstrations from distractors, an essential capability for real-world deployment.

**Dataset Partition and Benchmark Statistics.** Following established protocols, we first generate 5,000/2,000/1,000 questions for text-demo ICL/video-demo ICL and demonstration selection settings. To construct Demo-ICL-Bench, we then manually curate representative video demonstrations that highlight the role of demo-driven video in-context learning. Specifically, we sample 500 questions each for the text-demo and video-demo settings and 200 questions for the demonstration selection setting, resulting in a balanced benchmark of 1,200 questions in total.

### 3.3 LEARNING FROM IN-CONTEXT VIDEO DEMONSTRATIONS

We further train Demo-ICL models to validate the effectiveness of the proposed demo-driven video in-context learning task. The training pipeline is intentionally simple yet effective. As illustrated in Fig. 2 (ii), we adopt a two-stage strategy to progressively integrate demo-driven in-context learning. In the first stage, the model is fine-tuned on a tailored dataset to enhance fine-grained video comprehension and general in-context reasoning. In the second stage, we employ a customized DPO framework to specifically strengthen the model's capacity to learn from video demonstrations in context.

### 3.3.1 Video Supervised Fine-tuning

In this stage, our goal is to equip Demo-ICL with fine-grained video understanding and general in-context reasoning capabilities. To this end, we compile a large-scale dataset containing millions of samples drawn from diverse text–image pairs and video sources in open academic repositories. For image–text data, we rely on resources such as LLaVA-OneVision (Li et al., 2024a), VisualWebInstruct (Jia et al., 2025), and other widely used collections. For video data, we incorporate material from open-source projects including LLaVA-Video (Zhang et al., 2024c), Oryx (Liu et al., 2024), and Ola (Liu et al., 2025). To further enhance the model's ability for instructional video understanding, we additionally incorporate datasets such as COIN (Tang et al., 2019) and Cross-Task (Zhukov et al., 2019). We carefully exclude any videos that overlap with Demo-ICL-Bench to prevent data leakage and ensure fair evaluation. Finally, we perform subsampling on the generated dataset as described in Section 3.2 to explicitly introduce demo-driven video in-context learning signals during this stage. Together, these curated resources establish robust foundational capabilities and prepare the model for subsequent stages to enhance demo-driven video in-context learning.

### 3.3.2 Information-Assisted Direct Preference Optimization

Preference learning has become a critical component in the advancement of large language models, aiming to fine-tune outputs to better align with human preferences and improve real-world applicability. Traditional DPO algorithms generate multiple responses to the same query, after which a reward model ranks them and identifies preferred and rejected responses for training.

However, current models struggle with demo-driven video in-context learning, limiting their ability to generate high-quality responses. This limitation makes conventional DPO data construction pipelines less effective. To overcome these challenges, we propose an information-assisted DPO pipeline that integrates automatically generated assistive information, eliminating the need for manual annotation. For text-demo ICL tasks, we supply video timestamps to better align visual inputs with textual instructions, thereby improving accuracy. For video-demo ICL tasks, we pair video demonstrations with corresponding textual guidance to enhance response quality. For training, we define a preference dataset as $\mathcal{P} = \{(x^{(i)}, R_c^{(i)}, R_r^{(i)})\}_{i=1,\dots,|\mathcal{P}|}$, where each $x^{(i)}$ denotes the user request, and $R_c^{(i)}$ and $R_r^{(i)}$ represent the preferred and less preferred responses. We employ a reward model $r^*(x, y)$ to approximate preferences, and a higher score denotes a stronger preference. Following the approach introduced by (Rafailov et al., 2024), we can model the human preference distribution $p^*$ using the Bradley-Terry (BT) model (Bradley & Terry, 1952):

$$p^*(y_1 \succ y_2 \mid x) = \frac{\exp(r^*((x,I), y_1))}{\exp(r^*((x,I), y_1)) + \exp(r^*(x, y_2))} = \sigma(r^*((x,I), y_1) - r^*(x, y_2)), \quad (1)$$

where $I$ denotes the assistive information, and $\sigma$ denotes the logistic function. To estimate the parameters of the reward model, we can formulate the problem as a binary classification task and minimize the negative log-likelihood:

$$\mathcal{L}_R(r_\phi, \mathcal{P}) = -\mathbb{E}_{(x, R_c, R_r) \sim \mathcal{P}}[\log \sigma(r_\phi(x, R_c) - r_\phi(x, R_r))], \quad (2)$$

where $r_\phi$ is the reward model. This approach enables effective alignment with human preferences by allowing the model to use additional information that can be generated automatically, thus producing high-quality responses in an effective and scalable way. Using these responses as preferred outputs and treating normal responses as rejected, we perform multiple training rounds to obtain a sequence of models $\mathcal{M}_1, \dots, \mathcal{M}_T$, where each model $\mathcal{M}_{t+1}$ utilizes preference data $\mathcal{P}_t$ generated by the $t$-th model. Through the information-assisted DPO and iterative training strategy, we progressively endow Demo-ICL with strong demo-driven video in-context learning capabilities.

## 4 Experiments

We conduct extensive experiments on diverse video understanding benchmarks to assess the effectiveness of our proposed training strategies. First, we perform detailed evaluations on Demo-ICL-Bench, with results and analyses provided in Section 4.1. We then compare our method against state-of-the-art Video MLLMs on widely used benchmarks relevant to our tasks, as discussed in Section 4.2. Finally, we present essential analysis experiments in Section 4.3. These experimental results systematically

Table 1: **Evaluation results on Demo-ICL-Bench.** The benchmark assesses models across three tasks. For Text-demo ICL and Video-demo ICL, we report two types of accuracy: Demo. Acc (with demos) and w/o Demo(without demos), and the improvement ($\Delta_{ICL}$) attributed to demonstrations. DS refers to Demonstration Selection task. S.Acc refers to demonstration selection accuracy.

| Model | Size | Frame | Text-demo ICL | | | Video-demo ICL | | | DS | | Avg |
|---|---|---|---|---|---|---|---|---|---|---|---|
| | | | Demo. Acc | w/o Demo. | $\Delta_{ICL}$ | Demo. Acc | w/o Demo. | $\Delta_{ICL}$ | S.Acc | Acc | |
| Human | - | - | 84.0 | - | - | 80.4 | - | - | 88.0 | 76.0 | 80.1 |
| *Proprietary MLLMs* | | | | | | | | | | | |
| Gemini-2.5-Pro (Gemini Team, 2024) | - | - | 54.4 | - | - | 36.2 | - | - | - | 26.0 | 38.9 |
| GPT-4o (OpenAI, 2024) | - | - | 48.8 | - | - | 31.4 | - | - | - | 24.5 | 34.9 |
| *Open-Source Video MLLMs* | | | | | | | | | | | |
| Qwen2-VL (Wang et al., 2024a) | 7B | 32 | 29.0 | 21.8 | +7.2 | 22.4 | 24.0 | -1.6 | 38.0 | 14.5 | 22.0 |
| Ola (Liu et al., 2025) | 7B | 32 | 32.2 | 23.0 | +9.2 | 24.6 | 26.4 | -1.8 | 43.0 | 16.0 | 24.3 |
| LLaVA-Video (Zhang et al., 2024c) | 7B | 32 | 31.0 | 25.4 | +5.6 | 30.2 | 29.0 | +1.2 | 44.5 | 20.5 | 27.2 |
| Qwen2.5-VL (Bai et al., 2025) | 7B | 32 | 32.8 | 26.0 | +6.8 | 28.0 | 26.2 | +1.8 | 46.0 | 18.0 | 26.3 |
| InternVL-2.5 (Chen et al., 2025) | 8B | 32 | 29.2 | 21.8 | +7.4 | 27.4 | 27.8 | -0.4 | 42.5 | 18.0 | 24.9 |
| InternVL-3 (Zhu et al., 2025) | 8B | 32 | 31.4 | 26.6 | +4.8 | 27.0 | 26.4 | +0.6 | 44.0 | 16.5 | 25.0 |
| Video-R1 (Feng et al., 2025) | 7B | 32 | 33.6 | 27.6 | +6.0 | 27.4 | 26.6 | +0.8 | 48.0 | 17.5 | 26.2 |
| VideoChat-R1 (Li et al., 2025) | 7B | 32 | 34.4 | 27.0 | +7.4 | 28.2 | 26.8 | +1.4 | 52.0 | 18.5 | 27.0 |
| Qwen2.5-VL (Bai et al., 2025) | 72B | 32 | 45.0 | 24.2 | +20.8 | 25.6 | 25.2 | +0.4 | 54.0 | 18.0 | 29.5 |
| Demo-ICL (Ours, SFT) | 7B | 32 | 38.4 | 27.8 | +10.6 | 29.4 | 26.0 | +3.4 | 54.5 | 21.5 | 29.8 |
| Demo-ICL (Ours, DPO) | 7B | 32 | **43.4** | 29.4 | +14.0 | **32.0** | 27.6 | +4.4 | 58.0 | 24.0 | **33.1** |

validate the value of Demo-ICL-Bench and the advantages of our framework in using demo-driven video in-context learning for stronger alignment and generalization.

## 4.1 DEMO-ICL-BENCH

**Setup.** We evaluate both representative proprietary MLLMs and state-of-the-art open-source video MLLMs, reporting performance across three tasks. In addition to standard evaluations, we design experiments on text-demo and video-demo in-context learning tasks, including settings without explicit guidance, in order to better characterize the current capabilities and limitations of MLLMs in demo-driven video in-context learning. For proprietary models, we consider GPT-4o and Gemini-2.5-Pro. For open-source video MLLMs, we benchmark a diverse set of representative models, including InternVL-2.5 (Chen et al., 2025), InternVL-3 (Zhu et al., 2025), Qwen2-VL (Wang et al., 2024a), Qwen2.5-VL (Bai et al., 2025), LLaVA-Video (Zhang et al., 2024c), Ola (Liu et al., 2025), and Kimi-VL (Team & et al., 2025b). To capture the role of specialized video reasoning, we further include Video-R1 (Li et al., 2025) and VideoChat-R1 (Feng et al., 2025). Finally, to examine the effect of model capacity, we conduct experiments on both Qwen2.5-VL-7B and Qwen2.5-VL-72B.

**Text-demo In-context Learning.** As shown in Table 1, models perform poorly without demonstrations, indicating that in-context learning is essential for task success. When text demonstrations are provided, all models improve, demonstrating their ability to integrate task-specific knowledge from in-context text demonstrations. The extent of this improvement, however, strongly depends on model size: small models typically gain less than 10 points, whereas Qwen2.5-VL-72B improves by over 20 points, despite performing no better than smaller models without demonstrations. This highlights model scale as a critical factor for effective in-context learning. On Demo-ICL, the SFT model improves by over 10 points through targeted demonstration strategies, while the DPO model achieves state-of-the-art results among models of similar size. These results confirm that well-designed data curation, combined with preference-based training, substantially enhances generalization and efficiency in video in-context learning.

**Video-demo In-context Learning.** In the Video-demo ICL task, performance diverges from text-demo results: while some models extract information from video demonstrations, the gains are limited, and models such as InternVL-2.5, Qwen2-VL, and Ola even suffer degradation. This highlights the difficulty current MLLMs face in extracting and transferring temporal–visual cues for effective in-context learning. By contrast, Demo-ICL, equipped with demo-driven video in-context learning, consistently benefits from video demonstrations, though less pronounced compared to text-demo ICL tasks. Our findings indicate that dedicated strategies for video demonstrations are essential to narrow the gap between text and video guidance and to unlock further multimodal generalization.

**Demonstration Selection.** To approximate real-world scenarios where models must retrieve relevant demonstrations from large video pools, we evaluate them on the demonstration selection task. This

Table 2: **General Video Understanding.** Demo-ICL achieves superior performance on both general temporal understanding and knowledge acquisition tasks, demonstrating the effectiveness of the proposed demo-driven video in-context learning approach and the robustness of our training strategy.

| Model | Size | General Temporal Understanding | | | | Knowledge |
|---|---|---|---|---|---|---|
| | | VideoMME (wo / w sub) | MVBench | Long VideoBench | MLVU | VideoMMMU |
| *Proprietary Models* | | | | | | |
| GPT-4V (OpenAI, 2023b) | - | 59.9/63.3 | 43.7 | 49.2 | 59.1 | - |
| GPT-4o (OpenAI, 2024) | - | 71.9/77.2 | - | 66.7 | 66.7 | 61.2 |
| Gemini-1.5-Pro (Team & et al., 2024) | - | 73.2/79.8 | - | 64.0 | - | 70.4 |
| Gemini-2.5-Pro (Comanici & et al., 2025) | - | 84.3/86.9 | - | - | - | 83.6 |
| *Open-Sourced Video MLLMs* | | | | | | |
| VideoLLaMA2 (Cheng et al., 2024) | 7B | 47.9 / 50.3 | 54.6 | 36.0 | - | - |
| LLaVA-OneVision (Li et al., 2024a) | 7B | 58.2 / 61.5 | 56.7 | 56.3 | 64.7 | 33.9 |
| VideoLLaMA3 (Zhang et al., 2025a) | 7B | 66.2 / 70.3 | 69.7 | 59.8 | 73.0 | 47.0 |
| LLaVA-Video (Zhang et al., 2025b) | 7B | 63.3 / 69.7 | 58.6 | 58.2 | 70.8 | - |
| Qwen2.5-VL (Bai et al., 2025) | 7B | 65.1 / 71.6 | 69.6 | 56.0 | - | 47.4 |
| InternVL3.5 (Wang & et al., 2025) | 8B | 66.0 / 68.6 | 72.1 | 62.1 | 70.2 | - |
| LLaVA-Next-Video (Zhang et al., 2025b) | 34B | 52.0 / 54.9 | 70.2 | 50.5 | - | - |
| VILA-1.5 (Lin et al., 2024b) | 40B | 60.1 / 61.1 | - | - | 56.7 | 34.0 |
| VideoLLaMA2 (Cheng et al., 2024) | 72B | 61.4 / 63.1 | 62.0 | - | - | - |
| LLaVA-OneVision (Li et al., 2024a) | 72B | 66.2 / 69.5 | 59.4 | 61.3 | 66.4 | 48.3 |
| LLaVA-Video (Zhang et al., 2025b) | 72B | 70.5 / 76.9 | 64.1 | 61.9 | 74.4 | 49.7 |
| Qwen2.5-VL (Bai et al., 2025) | 72B | 73.3 / 79.1 | 70.4 | 60.7 | 74.6 | 60.2 |
| GLM-4.5V (Team & et al., 2025a) | 106B | 74.3 / 80.0 | 73.4 | 68.8 | 75.3 | 67.5 |
| Demo-ICL | 7B | 65.2 / 69.7 | 69.8 | 61.8 | 70.4 | 52.6 |

task assesses the ability to identify the correct reference video and answer the corresponding questions. We report both video selection accuracy and final question accuracy conditioned on the selected video. Results show that current models often struggle to capture global semantic information, leading to failures in retrieving appropriate demonstrations and producing a substantial gap from human performance. Existing approaches lack not only effective mechanisms for knowledge extraction and transfer, but also robust search and selection capabilities essential for demo-driven video in-context learning in real-world scenarios. Further analysis is provided in Section 4.3.

## 4.2 GENERAL VIDEO UNDERSTANDING

**Setup.** To evaluate the generalization ability of our Demo-ICL model, we conduct experiments on several widely used video benchmarks. Our analysis focuses on two main directions. First, we assess video knowledge acquisition using benchmarks such as VideoMMMU (Hu et al., 2025), a representative dataset designed to test how models acquire knowledge from videos. In this setting, the model must watch an entire video and answer questions based on its content, thereby evaluating its ability to learn, retain, and apply information in new contexts. This directly highlights the effectiveness of demo-driven video in-context learning, as the model uses demonstrations to generalize beyond the training distribution. Second, we evaluate on general temporal understanding benchmarks, including VideoMME (Fu et al., 2024a), MVBench (Li et al., 2024b), LongVideoBench (Wu et al., 2024), and MLVU (Zhou et al., 2024a), which target diverse tasks such as common video perception, action recognition, and long video understanding. Together, these benchmarks provide a comprehensive evaluation of Demo-ICL, covering both its demo-driven in-context learning generalization and its foundational video understanding capabilities.

**Results.** As shown in Table 2, Demo-ICL demonstrates competitive performance across all open-source MLLMs. On the knowledge acquisition benchmark VideoMMMU, Demo-ICL performs on par with recently released models of comparable size and even surpasses some larger counterparts. These results highlight not only the strength of our model in visual reasoning but also the effectiveness of demo-driven video in-context learning as a paradigm for scalable knowledge acquisition. By using demonstrations directly within the input context, Demo-ICL generalizes beyond memorized content, suggesting a promising direction toward more flexible and human-like video understanding. Moreover, on general video understanding benchmarks, Demo-ICL achieves performance comparable to newly released models of similar size, indicating that the proposed demo-driven ICL mechanism can be seamlessly integrated without compromising common temporal understanding, while at the same time enhancing knowledge acquisition. Our findings provide strong evidence that demo-driven video ICL offers a scalable and robust path toward advancing video-based reasoning and understanding.

### 4.3 ANALYSIS EXPERIMENTS

In this section, we present detailed analyses focusing on the challenges of video-demo in-context learning and strategies for training an effective demo-driven model. We highlight the limitations of current models and demonstrate the effectiveness of information-assisted DPO.

**Why is the Video-demo ICL task challenging?** We provide deeper insights into why the Video-demo ICL task poses significant challenges for current MLLMs, with results summarized in Table 3. First, we test Demo-ICL with more densely sampled frames, and the improvements demonstrate that fine-grained visual cues are critical for demo-driven video in-context learning. Using 128 frames, we further conduct an experiment where the reference video is identical to the query video, thus providing the model with the full content as context. The performance gains in this setting suggest that direct grounding and perception are far easier than knowledge transfer through in-context demonstrations, as the model can process visuals effectively

Table 3: Ablation study on evaluation settings.

| Settings | Video-demo ICL |
|---|---|
| Base(32 frames) | 29.4 |
| 128 frames | 30.4 |
| +Repeat Video | 38.6 |
| +Reference Clips | 35.8 |
| +ASR & Captions | **45.4** |

but struggles to adapt that knowledge to new scenarios. We further evaluate the use of reference clips as contextual demonstrations, where only the segments depicting the immediate next-step action are provided as in-context examples. This setting reveals a fundamental challenge for Video-demo ICL: models struggle to accurately align and match temporal evidence across demonstrations. Moreover, replacing clips with ASR transcripts and captions yields additional improvements, revealing that current MLLMs still lack robust fine-grained video comprehension and often fail to abstract or summarize clips into precise knowledge for reasoning and further adaptation.

Taken together, these findings highlight why video-demo ICL is uniquely challenging: it requires not only perception but also temporal alignment, abstraction, and flexible knowledge transfer. This underscores the need for models that can truly leverage demonstrations as dynamic sources of contextual information, a critical capability for advancing video understanding and reasoning.

**How to train a good demo-driven video in-context learning model?** We perform ablation studies to assess the effectiveness of our training strategies, with the results summarized in Table 4. The findings indicate that incorporating instructional videos allows Demo-ICL to leverage in-context demonstrations and adapt to novel scenarios,

Table 4: Training setting ablations.

| Settings | Text-ICL | Video-ICL | DS | Avg |
|---|---|---|---|---|
| w/o Instructional Videos | 34.0 | 26.2 | 19.0 | 26.4 |
| Demo-ICL (SFT) | 38.4 | 29.4 | 21.5 | 29.8 |
| Vanilla DPO | 40.0 | 30.0 | 22.0 | 30.7 |
| Demo-ICL (DPO) 1-round | 41.8 | 30.8 | 22.5 | 31.7 |
| Demo-ICL (DPO) | **43.4** | **32.0** | **24.0** | **33.1** |

yielding significant performance improvements on Demo-ICL-Bench. These results emphasize the importance of high-quality instructional data in enabling models to generalize beyond basic perception and toward deeper contextual video understanding.

We further investigate the impact of training algorithms. When trained with vanilla DPO, the model struggles to produce high-quality responses, yielding noisy data pairs and only marginal improvements. By contrast, our information-assisted DPO method provides richer feedback signals, which significantly enhance response quality and overall performance. Through iterative training strategy, Demo-ICL gradually learns to learn through in-context demonstrations, finally reach superior performance. These comparisons reveal that both the quality of demo-driven video data and the design of training strategies are essential for effective video ICL. Together, these results indicate that building a strong video in-context understanding model requires not only carefully structured demonstrations but also training paradigms that use contextual information.

## 5 CONCLUSION

In this paper, we introduce a novel task, Demo-driven Video In-Context Learning, which focuses on learning from in-context instructional demonstrations. To facilitate evaluation, we present Demo-ICL-Bench, a benchmark consisting of 1,200 challenging questions designed to assess demo-driven video in-context learning capabilities. To effectively address this task, we further propose Demo-ICL, a video MLLM equipped with enhanced in-context learning abilities. Extensive experiments reveal that existing MLLMs struggle with Demo-driven ICL, whereas Demo-ICL overcomes these challenges, achieving superior video understanding and in-context knowledge acquisition capabilities, thereby paving the way for future advancements.

ETHICS STATEMENT

This research strictly follows the ICLR Code of Ethics. We emphasize that Demo-ICL-Bench, the Demo-ICL model, and the collected dataset are entirely non-commercial, with their development carefully avoiding any ethical or legal issues, particularly concerning intellectual property. Our methodology rigorously upholds copyright integrity through two protective measures: (1) all foundational project descriptions are the original creations of the authors; (2) we strictly adhere to the licenses of collected videos, ensuring no legal issues arise from data collection or release.

REPRODUCIBILITY STATEMENT

To ensure the reproducibility of our results, we have made considerable efforts to provide the necessary details and materials. Specifically, we have included a comprehensive description of the dataset creation process in Section 3.2. More implementation details about model training and data collection details are described in detail in Section B.

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

APPENDIX

We provide supplementary documents to support our research. The details of Large Language Model usage are presented in Section A. Implementation details are outlined in Section B. Additional visualization results are presented in Section C, followed by further experimental analysis in Section D. We also provide a more comprehensive discussion of related work in Section E. Finally, we discuss the limitations of our work in Section F.

## A    LARGE LANGUAGE MODEL USAGE

In this paper, we clarify that large language models (LLMs) are employed solely to support and refine the writing process. Specifically, we use LLMs to provide sentence-level suggestions and to enhance the overall fluency of the text.

## B    IMPLEMENTATION DETAILS

### B.1    EXPERIMENT DETAILS

In this section, we detail the implementation of Demo-ICL. The Demo-ICL model is built upon Ola-Video, a highly pretrained multimodal understanding model that integrates OryxViT as its visual encoder to process native arbitrary-resolution visual inputs, alongside Qwen2.5 as the language model. For the training process, we construct a customized dataset to establish foundational image and video understanding capabilities. For image data, resolutions range from 768 to 1536, while for video data, the number of frames is capped at 64, with frame resolutions varying between 288×288 pixels and 480×480 pixels. During training, the maximum token length is set to 16,384, and a learning rate of 1e-5 is used throughout both stages. In the DPO (Direct Preference Optimization) training phase, we curate 5,000 samples using the specified pipeline and apply a learning rate of 5e-7. A batch size of 256 is maintained across both fine-tuning stages and the DPO phase, with experiments conducted using 64 NVIDIA A800 GPUs.

### B.2    DATA COLLECTION DETAILS

In the data generation process, we utilize Qwen2.5-72B as our LLM and Qwen2.5-VL-72B as our MLLM within the pipeline. For generating text instructions, we first use Qwen2.5-72B to create summarized instruction steps. Then, when refining these steps with the MLLM, we forward each step along with 64 uniformly sampled frames from the corresponding video clips. For generating questions for video-demo ICL, we provide the text instructions of paired videos and ask the LLM to assess their reasonableness for question generation. Both the LLM and MLLM are deployed using four NVIDIA A800 GPUs.

## C    VISUALIZATIONS

We present visualization results to clarify the task design of Demo-ICL-Bench. These results are shown in Fig. 3 and Fig. 4.

## D    MORE ANALYSIS EXPERIMENTS

### D.1    GENERAL VIDEO UNDERSTANDING ON VIDEO-MME

We further evaluate the Demo-ICL model on general video understanding tasks of varying lengths and scenarios. Specifically, we employ the VideoMME benchmark to highlight its offline video comprehension capabilities, providing a broader assessment beyond domain-specific settings.

**Setup.**    To further evaluate the generalization ability of Demo-ICL on diverse video understanding tasks, we adopt the Video-MME benchmark (Fu et al., 2024a). The dataset consists of 900 videos (254 hours) covering 6 visual domains and 30 subfields, with durations ranging from 11 seconds

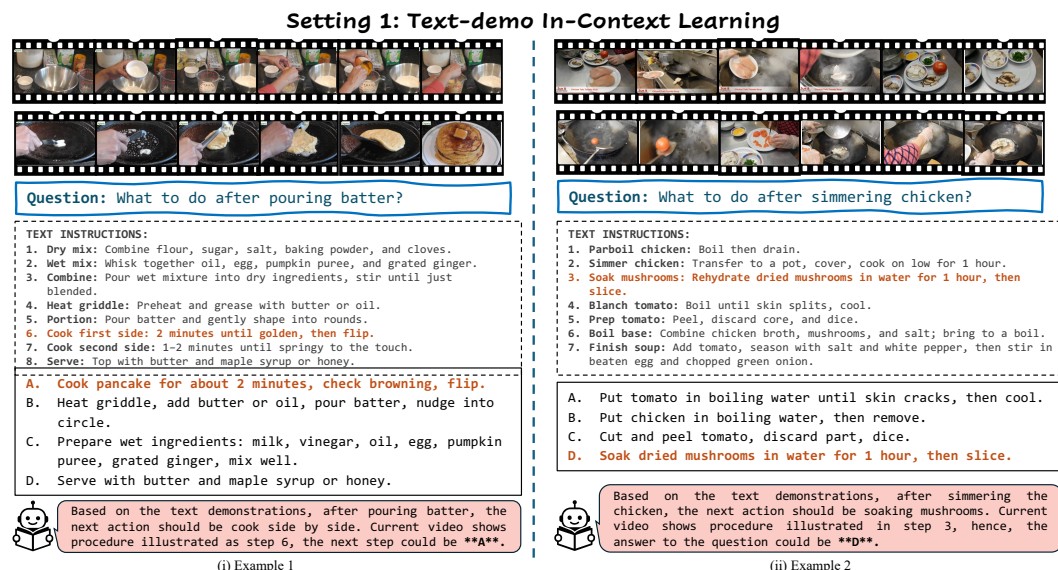

Figure 3: **Visualization of Text-demo In-Context Learning.** This figure provides 2 examples to illustrate the text-demo in-context learning task, where the text instructions will be provided along with the target video as the inputs.

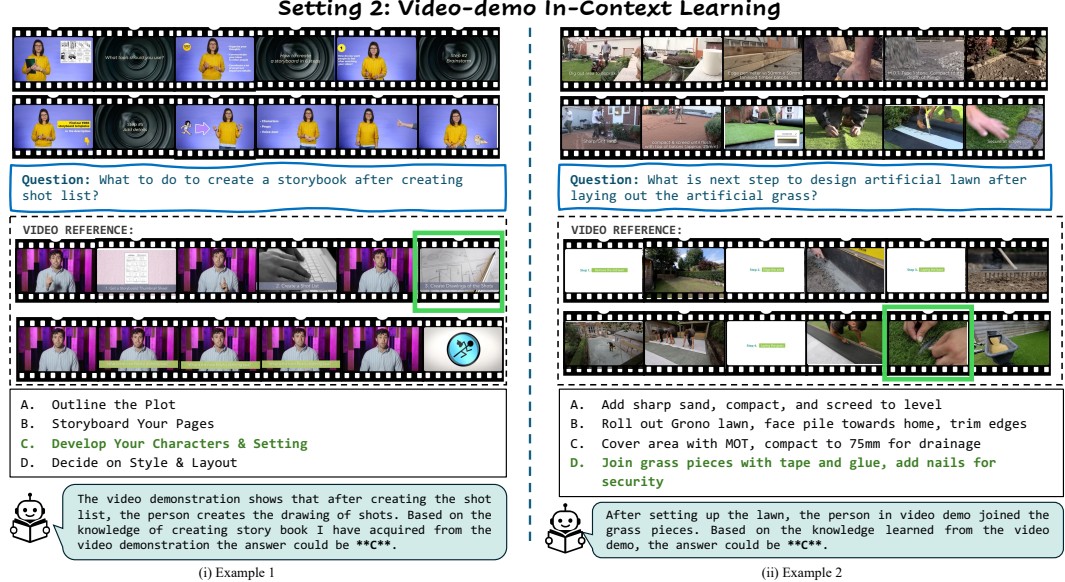

Figure 4: **Visualization of Video-demo In-Context Learning.** This figure provides 2 examples to illustrate the video-demo in-context learning task, where a video demonstration will be provided together with the target video input.

Table 5: Performance of Demo-ICL compared to previous MLLMs on Video-MME across short, medium, and long durations, under without "subtitles" and with "subtitles" settings.

| Models | LLM Params | Short (%) | | Medium (%) | | Long (%) | | Overall (%) | |
|---|---|---|---|---|---|---|---|---|---|
| | | w/o subs | w/ subs | w/o subs | w/ subs | w/o subs | w/ subs | w/o subs | w/ subs |
| *Commercial MLLMs* | | | | | | | | | |
| GPT-4V (OpenAI, 2023a) | - | 70.5 | 73.2 | 55.8 | 59.7 | 53.5 | 56.9 | 59.9 | 63.3 |
| GPT-4o (OpenAI, 2024) | - | 80.0 | 82.8 | 70.3 | 76.6 | 65.3 | 72.1 | 71.9 | 77.2 |
| Gemini 1.5 Flash (Gemini Team, 2024) | - | 79.7 | 83.6 | 68.4 | 74.7 | 61.1 | 68.8 | 70.3 | 75.0 |
| Gemini 1.5 Pro (Gemini Team, 2024) | - | 81.7 | 84.5 | 74.3 | 81.0 | 67.4 | 77.4 | 75.0 | 81.3 |
| *Open-source Video MLLMs* | | | | | | | | | |
| LongVA (Zhang et al., 2024a) | 7B | 61.1 | 61.6 | 50.4 | 53.6 | 46.2 | 47.6 | 52.6 | 54.3 |
| VITA 1.5 (Fu et al., 2025) | 7B | 67.0 | 69.9 | 54.2 | 55.7 | 47.1 | 50.4 | 56.1 | 58.7 |
| mPLUG-Owl3 (Ye et al., 2024) | 7B | 70.0 | 72.8 | 57.7 | 66.9 | 50.1 | 64.5 | 59.3 | 68.1 |
| TimeMarker (Chen et al., 2024a) | 8B | 71.0 | 75.8 | 54.4 | 60.7 | 46.4 | 51.9 | 57.3 | 62.8 |
| MiniCPM-V 2.6 (Yao et al., 2024) | 8B | 71.3 | 73.5 | 59.4 | 61.1 | 51.8 | 56.3 | 60.9 | 63.7 |
| VILA-1.5 (Lin et al., 2024b) | 34B | 68.1 | 68.9 | 58.1 | 57.4 | 50.8 | 52.0 | 59.0 | 59.4 |
| Oryx-1.5 (Liu et al., 2024) | 34B | 77.3 | 80.6 | 65.3 | 74.3 | 59.3 | 69.9 | 67.3 | 74.9 |
| Qwen2-VL (Wang et al., 2024a) | 72B | 80.1 | 82.2 | 71.3 | 76.8 | 62.2 | 74.3 | 71.2 | 77.8 |
| LLaVA-Video (Zhang et al., 2024b) | 72B | 81.4 | 82.8 | 68.9 | 75.6 | 61.5 | 72.5 | 70.6 | 76.9 |
| Demo-ICL | 7B | 78.6 | 79.1 | 63.9 | 68.8 | 53.2 | 61.1 | 65.2 | 69.7 |

to 1 hour, categorized into Short, Medium, and Long. In addition to visual content, VideoMME provides audio and subtitles, enabling a multimodal and comprehensive evaluation of video MLLMs. Under this setting, the model is required to watch an entire video and then answer corresponding questions, which allows us to systematically assess robustness across varying durations, modalities, and domains, in comparison with both open-source and commercial MLLMs.

**Results.** Table 5 summarizes the overall performance of Demo-ICL across short, medium, and long video tracks. Demo-ICL achieves strong results on all three tracks, demonstrating robust capabilities across different temporal lengths. It surpasses open-source video MLLMs with similar parameter sizes (7B), achieves comparable results to larger models (34B), and competes closely with some commercial MLLMs. Notably, on long-duration videos, which pose greater challenges due to extended temporal dependencies, Demo-ICL demonstrates its long video understanding capabilities, maintaining consistent performance over time.

# E MORE DISCUSSION ON RELATED WORKS

In this section, we will include more details of related works.

**Multimodal Video Understanding for Knowledge Acquisition.** Recent research in video understanding has moved beyond low-level perception towards extracting structured knowledge from videos, like procedural steps, events, and concepts. Large-scale instructional datasets have been instrumental in this shift. For example, as mentioned in 2, a lot of instructional datasets (Miech et al., 2019; Tang et al., 2019; Zhukov et al., 2019) have driven the development of models that seek to learn high-level knowledge from video, rather than just recognize objects or actions. Moreover, VidSitu (Sadhu et al., 2021) addresses video situation recognition by densely annotating 10-second movie clips with semantic role labels, which provides a symbolic knowledge representation of the video. By learning to predict such structured representations, models can acquire a form of event knowledge from videos. Similarly, HT-Step (Afouras et al., 2023) aligns the textual instructions from wikiHow (Koupaee & Wang, 2018) with corresponding segments in instructional videos. It provides 116k temporal segment annotations in 20k how-to videos, each labeled with a step description from wikiHow, enabling models to learn to ground declarative knowledge in procedural video footage.

To better learn from such knowledge-intensive data, early multimodal learning approaches applied language-modeling techniques to video data. For example, VideoBERT (Sun et al., 2019) quantizes video frames into discrete "visual words" and then uses a BERT-like transformer to learn joint representations of sequences of visual tokens and narration text. Following models such as ActBERT (Zhu & Yang, 2020) extended this masked language modeling paradigm to action recognition data, and ClipBERT (Lei et al., 2021) improved efficiency by sampling sparse key frames for end-to-end video-text pretraining. By learning from millions of narrated video clips, these models demonstrate

an ability to embed procedural and commonsense knowledge implicitly in their representations. Zhou et al. (2023) proposed the model Paprika used PKG-based pre-trainng procedure to generate psuedo labels for instructional video to train. StepFormer (Dvornik et al., 2023) addresses the problem of discovering and localizing key procedure steps in instructional videos without human supervision. It uses video with subtitles (ASR) only, with a transformer decoder that attends to video frames via learnable queries to produce a sequence of key steps. Chen et al. (2024b) proposes a framework MPTVA, that aligns video segments with procedure steps derived via LLM from narration text via long-term semantic similarity and short-term fine-grained similarity.

Table 6: **Related Work for Demo-ICL-Bench.** Demo-ICL-Bench stands out due to its demo-driven video in-context learning settings, setting it apart from previous video benchmarks.

| Benchmark | Video Domain | #Videos | #QAs | Video-ICL | Annotation |
|---|---|---|---|---|---|
| ActivityNet-QA (Fabian Caba Heilbron & Niebles, 2015) | Human Activities | 800 | 8000 | ✗ | Manual |
| How2QA (Li et al., 2020) | Instructional Videos | 1166 | 2852 | ✗ | Manual |
| KnowIT-VQA (Garcia et al., 2020) | TV Show | 207 | 24k | ✗ | Manual |
| NExT-QA (Xiao et al., 2021) | Web Videos (Causal/Temporal) | 5.4k | 52k | ✗ | Manual |
| MVBench (Li et al., 2024b) | Benchmark Videos | 3641 | 4000 | ✗ | Auto |
| VideoMME (Fu et al., 2024b) | YouTube Videos | 900 | 2700 | ✗ | Manual |
| VideoMathQA (Rasheed et al., 2025) | Instructional Videos | 420 | 420 | ✗ | Manual |
| VideoMMMU (Hu et al., 2025) | Lectures | 300 | 900 | ✗ | Manual |
| Demo-ICL-Bench | Instructional Videos | 1200 | 1200 | ✓ | Mixed |

**Multimodal In-Context Learning.** Inspired by the textual CoT prompting, recent works curate multimodal datasets with human-written rationales to encourage step-by-step prompting. Video-CoT (Wang et al., 2024b) provides video QA examples paired with detailed explanations, while Video-Espresso (Han et al., 2025) scales this approach to large collections of reasoning exemplars. Beyond data-centric methods, Arnab et al. (2025) propose Temporal Chain-of-Thought, an inference strategy for long videos where the model iteratively selects relevant clips and reasons over them, enabling efficient multi-step reasoning over extended sequences. A complementary line of work extends retrieval-augmented generation (RAG) to video. VideoRAG (Ren et al., 2025) and related work (Tevissen et al., 2024) index long videos into databases of visual and textual descriptors. At query time, relevant segments and transcripts are retrieved and passed to the language model as context, grounding answers in explicit video evidence. This improves factual accuracy, transparency, and scalability, especially for long videos where direct end-to-end processing is infeasible.

# F  LIMITATIONS AND FUTURE DIRECTIONS

In this section, we discuss the limitations of our work. The Demo-ICL model does not include a specialized architecture for demo-driven video in-context learning. Instead, we employ a customized training strategy to achieve this functionality. Our goal is to equip current MLLMs with demo-driven video in-context learning capability without requiring architectural modifications, thereby simplifying the integration of these new capabilities and the maintenance of previous multimodal understanding.

Additionally, we did not explore how models can effectively learn from diverse contexts, such as different modalities or resources. This ability is more similar to the natural human learning process, where individuals can draw on a wide range of resources, such as text instructions and instructional videos, to enhance understanding simultaneously. Combining various types of contextual information to improve in-context learning and ultimately enhance a model's performance on new tasks remains a significant challenge.

