# OpenReview forum: "Demo-ICL: In-Context Learning for Procedural Video Knowledge Acquisition"
_ICLR.cc/2026/Conference — ICLR 2026 Conference Withdrawn Submission_

### Official Review · Reviewer_96Gh · 2025-10-27

**Soundness:** 2
**Presentation:** 1
**Contribution:** 2
**Rating:** 2
**Confidence:** 3

**Summary:**

The paper introduces Demo-ICL, a framework for demo-driven video in-context learning. It aims to enable multimodal large language models to learn procedural knowledge through in-context demonstrations. The work also proposes Demo-ICL-Bench, a new benchmark derived from instructional video data (mainly from HowTo100M), designed to test a model’s ability to learn from demonstrations in both text and video formats. The model combines video-supervised fine-tuning with information-assisted direct preference optimization to improve multimodal adaptation and generalization.

**Strengths:**

1. The task of demo-driven video in-context learning is conceptually interesting and relevant for next-generation multimodal reasoning, particularly for applications like robotics or educational AI systems.
2. The benchmark covers three distinct tasks: text-demo, video-demo, and demonstration selection.

**Weaknesses:**

1. The claims are overstated — particularly the assertion that Demo-ICL “unveils future research directions” and “demonstrates superior knowledge acquisition.” The performance margins are relatively small and inconsistent across subtasks.
2. Several sentences and sections are unclear (e.g., descriptions of how demonstrations are chosen and how textual vs. video modalities interact). It’s often difficult to follow the flow between task setup, dataset construction, and evaluation.
3. The abstract and introduction are confusing and fail to clearly communicate the model’s novelty compared to existing multimodal ICL paradigms.
4. The paper refers to “recent” progress, yet most citations stop at 2023. There’s little awareness of 2024–2025 works on multimodal retrieval and contextual grounding (e.g., VideoRAG, Chain-of-Thought for Video).
5. The central message and motivation are obscured by excessive text and lack of focus on specific contributions.
6. (from note 6) The method described on page 2 is essentially retrieval-based, yet the paper does not discuss its relation to multimodal retrieval systems or retrieval-augmented generation frameworks.
7. The use of LLMs to process noisy web data (HowTo100M) is problematic: large models cannot resolve multimodal misalignments using text alone. There is no verification (e.g., user studies, human annotations) of the resulting “summaries.” based on ASRs only.
8. The evaluation lacks comparison to few-shot or multimodal adaptation baselines such as CoOp (IJCV 2022), ViFi-CLIP (CVPR 2023), and newer 2024–2025 models using contextual tuning.
9. Many sentences are hard to interpret (e.g., “from video” or “based on demonstration”), making it unclear whether the model is trained to answer about or within the video context.
10. The difference between tasks (text-demo vs. video-demo) is poorly delineated; the text sources in “1” are not explained clearly, and Figure 2 provides little clarification.
11. Task robustness and evaluation methodology are underdefined — it’s unclear how the model handles the same tasks executed in different orders across videos.
12. The term “established protocols” (Section 3.2) is mentioned but not explained or referenced.
13. Table 2 does not show improvements compared to same-size models, and the base model (Ola) is not evaluated for comparison.
14. The utility of the proposed method is questionable given marginal quantitative improvements and limited qualitative evidence.
15. The training and evaluation datasets are not sufficiently detailed; while sources like HowTo100M are described, no concrete statistics are provided.

**Questions:**

see above

---

### Official Review · Reviewer_xWfV · 2025-10-30

**Soundness:** 3
**Presentation:** 3
**Contribution:** 3
**Rating:** 4
**Confidence:** 3

**Summary:**

The paper introduces Demo-driven Video In-Context Learning (Demo-ICL), a novel task that challenges MLLMs to learn procedural knowledge from instructional videos or text demonstrations and apply it to answer questions about new target videos. To benchmark this capability, the authors propose Demo-ICL-Bench, a curated dataset of 1,200 instructional YouTube videos with associated questions, supporting three task settings: text-demo ICL, video-demo ICL, and demonstration selection. Unlike existing video benchmarks that rely on static or internal knowledge, Demo-ICL emphasizes dynamic, example-based learning. The authors also present Demo-ICL, a 7B-parameter MLLM trained via a two-stage pipeline—video-supervised fine-tuning and information-assisted DPO—to enhance its ability to learn from in-context demonstrations. Extensive experiments show that Demo-ICL significantly outperforms both proprietary and open-source models on Demo-ICL-Bench and general video understanding benchmarks like VideoMMMU and VideoMME, highlighting its effectiveness in procedural knowledge acquisition and temporal reasoning.

**Strengths:**

1. The paper invents a new learning paradigm—demo-driven video in-context learning—by reframing video understanding as “watch a tutorial, then perform the next step in a different video”. This moves the community goalpost from static recognition to rapid procedural knowledge uptake, an angle no prior benchmark or method has systematically targeted.
2. Demo-ICL-Bench is built with industrial-grade rigor: 1,200 YouTube instructional videos filtered for language, length, and narrated step boundaries; ASR timestamps paired with LLM-distilled step lists; human review to guarantee answerability; and a three-task split (text demo, video demo, demonstration retrieval) that stress-tests complementary skills.

**Weaknesses:**

1. In the third task the model must first retrieve the correct tutorial from a 200-video pool and then answer. The paper only reports top-1 selection accuracy (S-Acc) and final QA score, but never analyzes why retrieval fails.
2. Evaluation metric blind to step-granularity. Accuracy is binary on the next step; it gives no partial credit when the model predicts “add onion” instead of “add onion and garlic.” A softer metric such as  human preference win-rate would reward models that learn coarse procedure structure even when fine details differ.
3. The paper does not clarify whether the model’s final accuracy stems from the large model’s built-in prior knowledge or from genuine learning within the provided demonstration videos.

**Questions:**

Please refer to weaknesses

---

### Official Review · Reviewer_Eprt · 2025-10-31

**Soundness:** 2
**Presentation:** 3
**Contribution:** 3
**Rating:** 6
**Confidence:** 4

**Summary:**

This paper introduces Demo-driven Video In-Context Learning (Demo-ICL), a new task that evaluates how Multimodal Large Language Models (MLLMs) can learn and adapt from in-context demonstrations instead of relying solely on static knowledge. To benchmark this ability, the authors construct Demo-ICL-Bench, comprising 1,200 instructional YouTube videos with both text and video-based demonstrations. They further propose Demo-ICL, an MLLM trained with a two-stage pipeline—video-supervised finetuning and information-assisted direct preference optimization—to better utilize contextual examples. Experiments on state-of-the-art MLLMs validate the benchmark’s difficulty and show that Demo-ICL achieves superior adaptability and understanding.

**Strengths:**

1. The paper is well-written and easy to follow.
2. The motivation is strong — exploring In-Context Learning for video understanding is both important and underexplored, addressing a genuine gap in current MLLM research.
3. The work is comprehensive, presenting not only a new task formulation but also a dedicated benchmark and a well-designed method to address it.
4. The experimental evaluation is thorough, including results on both the proposed Demo-ICL-Bench and general benchmarks, and providing insightful analysis of the conditions and capabilities required for effective video-demo ICL.

**Weaknesses:**

1. The questions in the benchmark appear overly detailed, which may reduce the contribution of the target video itself. I'm very curious about the performance of the Text-demo In-Context Learning subtask without the target video (i.e., question-only accuracy).
2. The paper lacks detailed explanation and examples for the Demonstration Selection subtask. It is unclear whether each video is accompanied by additional information, such as subtitles or ASR transcripts.
3. Beyond providing a new evaluation dimension for MLLMs, the practical significance of the Demo-ICL task and the well-trained Demo-ICL model is not clearly demonstrated. For example, can it serve as an effective agent in certain real-world video understanding scenarios?
4. The evaluation section omits key details. Although Demo-ICL is trained based on the Ola-Video, Tables 1 and 2 do not include Ola-Video results, making it impossible to assess whether Demo-ICL training retains general capabilities. Moreover, it is unclear whether the results in Table 2 correspond to the SFT or DPO version of the model—as in my experience, DPO fine-tuning can often degrade general performance substantially.

**Questions:**

Please refer to the Weaknesses section. Addressing these issues would significantly strengthen the paper and likely improve my score.

---

### Note · Authors · 2025-11-12

I have read and agree with the venue's withdrawal policy on behalf of myself and my co-authors.